# Parameter Monte Carlo Tree Search: Efficient Chip Placement via Transfer Learning

## Abstract

Automated chip placement is an important problem in enhancing the design and effectiveness of computer chips. Previous approaches have employed transfer learning to adapt knowledge obtained via machine learning from one chip placement task to another. However, these approaches have not notably reduced the necessary chip design time, which is crucial for minimizing the total resource utilization. This paper introduces a novel transfer learning approach called Parameter Monte Carlo Tree Search (PMCTS) that utilizes MCTS to transfer the learned knowledge from deep reinforcement learning (RL) models trained on one chip design task to another chip design by searching directly over the model parameters to generate models for efficient chip placement. We employ MCTS to escape the local optima reached by training from scratch and fine-tuning methods. We evaluate our methodology on four chip design tasks from the literature: Ariane, Ariane133, IBM01, and IBM02. Through extensive experiments, we find that our approach can generate models for optimized chip placement in less time than training from scratch and fine-tuning methods when transferring knowledge from complex chip designs to simpler ones.

## 1 Introduction

Chip placement is a fundamental and time-consuming stage in the chip design process (Cheng et al., 2022). The process involves placing the various components, such as macros and standard cells, of a netlist file, which contains a logical description of a circuit's interconnections, in specific positions on the chip layout (Ellis-Monaghan & Gutwin, 2003). Standard cells refer to the fundamental logic gates such as NAND, NOR, and XOR, while macros represent functional blocks like static random-access memory (SRAM). The goal of placement is to minimize power, performance, and area (PPA) metrics while adhering to restrictions like placement density and routing congestion (Cheng & Yan, 2021). This is a longstanding problem, but the exponential growth of artificial intelligence (AI) technology has generated an increased demand for advanced computational hardware. However, due to the end of Moore's Law and Dennard scaling, it is essential to shift towards specialized architecture to cope with the rapidly increasing computational requirements of AI (Mirhoseini et al., 2020). As such, AI-based techniques are expected to accelerate the chip design cycle, establishing a mutually beneficial connection between hardware and AI.

Transfer learning methods can prove beneficial in reducing the cost and resource requirements of developing machine learning (ML) models from scratch for novel domains (You et al., 2023). These methods utilise pre-existing model weights rather than random initialization to reduce the computing demand and enable faster adaptation, which in turn can support more design iterations (Cao et al., 2010). Many transfer learning approaches exist, including zero-shot learning (Xian et al., 2017) and domain adaptation (Farahani et al., 2021) to transfer knowledge from a source domain to a target domain. However, zero-shot learning often requires the class sets or features to be identical, and in the case of domain adaptation approaches, a degree of parallelism between the domains is necessary. As such, for chip design tasks, fine-tuning has been the most common approach Mirhoseini et al. (2020), though they have not demonstrated the expected increases in speed.

Search algorithms have played a crucial role in domains such as Neural Architecture Search (NAS) (Li & Peng, 2020) and Hyper Parameter Optimization (HPO) (Alibrahim & Ludwig, 2021). These approaches are less commonly applied directly to the parameters of a model (Singamsetti et al.,

2021; Mahajan & Guzdial, 2022; Doosti & Guzdial, 2023). Building on prior search-based transfer learning approaches, we propose an approach for chip placement that directly updates network parameters via search.

Our study introduces a novel transfer learning method based on Monte Carlo Tree Search (MCTS) and MCTS using Upper Confidence Bounds for Trees (UCB) to balance exploration with exploitation (Kocsis & Szepesvári, 2006; Auer et al., 2002). MCTS is probabilistic and heuristic-driven, allowing it to adapt to the most promising search regions (Deng & Wu, 2023) and guiding our proposed approach to effectively explore inside promising search regions. Our approach aims at directly optimizing model parameters within a complex search space. It iteratively updates a search tree by simulating possible outcomes for an RL agent in the chip design environment. Every node in the tree corresponds to two networks: a policy network and a value network, and each node is associated with a cost value: the performance of these networks in a specific chip design environment. The policy network, denoted as $\pi_\theta(a|s)$, determines the probability of selecting an action $a$ in a given state $s$, with the parameter $\theta$ influencing this behaviour, and the value network provides an estimation of the expected reward for the chip placement. The cost value is the weighted total of wirelength, congestion, and density costs associated with the chip design task; a node is considered more efficient if it creates chip placements with less cost value.

The contributions of this paper are summarized as follows:

- We propose a novel transfer learning-based optimization approach named Parameter Monte Carlo Tree Search (PMCTS) utilizing the MCTS algorithm to dynamically update the parameters of a neural network to adapt models with a relatively small amount of computation and which can generate efficient chip placements.
- We validate the proposed methodology across four distinct chip design tasks—Ariane, Ariane133, IBM01, and IBM02—highlighting its effectiveness and stability in diverse chip design tasks.
- We show that transferring knowledge from a complex chip design task, such as IBM02, to a simpler chip design, like Ariane, outperforms both training from scratch and fine-tuning approaches in terms of creating chip placements with reduced cost value in lesser timeframes. However, transferring knowledge from a simpler chip design to a more complex one does not yield the same benefits.

## 2 RELATED WORK

This section covers related work in transfer learning, parameter search, and chip placement problems.

### 2.1 TRANSFER LEARNING

Transfer learning using deep neural networks (DNNs) refers to the process of transferring knowledge and parameters from a DNN trained on a source dataset to another DNN aiming to address a similar target problem (Shafahi et al., 2019). Several methods, including zero-shot (Xian et al., 2017), one-shot (Fei-Fei et al.), and few-shot (Ravi & Larochelle, 2017) approaches, exist to transfer knowledge from a source domain to a target domain. However, these approaches often rely on manually authored features or significant additional data to effectively guide the knowledge transfer process. Our proposed approach does not require the incorporation of additional manually authored features or significant additional training to adapt to a target domain.

### 2.2 PARAMETER OPTIMIZATION

Hyperparameter Paramater Optimization (HPO) is the most common search problem associated with DNNs (Feurer & Hutter, 2019). However, our primary focus is on optimizing parameters rather than hyperparameters.

The majority of current neural network (NN) research has been on deep learning, where the primary approach for training NNs is backpropagation (Rumelhart et al., 1986), an algorithm used to compute the gradient of the loss function. Backpropagation-based approaches have been useful in

different domains for determining optimal parameters. However, backpropagation can often become stuck in local minima (Siddique & Tokhi, 2001). Further, the optimal parameters for backpropagation appear to differ depending on the specific scenario. Sparse regularization methods in NNs focus on a subset of the network to enhance computational and space efficiency without compromising performance by addressing the issue of redundant and correlated structures. The backpropagation technique proposed by Sun et al. (2017) involves computing a very small but vital part of the gradient and modifying only the corresponding small portion of the parameters in each update. This approach results in highly sparsified gradients that only modify highly relevant parameters for the given training sample. Additionally, pruning strategies may compress a model without significantly reducing its capacity for prediction (Yeom et al., 2021). Our approach can be understood as a type of sparse optimization, where we rely on search to identify parameters for optimization.

Neuroevolution stands as an alternate optimization approach that involves the training of NNs using evolutionary algorithms. Making use of evolutionary search facilitates significant functionalities that are not commonly available to gradient-based methodologies, such as the ability to learn neural network's activation functions and architectures (Stanley et al., 2019). Aly et al. (2019) introduce an evolutionary metaheuristic aimed at optimizing DNNs. The objective is to search multiple regions of the search space while preserving a specific distance between those regions to guarantee diversity. Zhou et al. (2024) introduce a neuroevolutionary diversity policy search method for multi-objective reinforcement learning (RL) challenges, utilizing gradient-based genetic operations to equip each policy with a buffer for gathering past experiences. A deep genetic algorithm proposed by Such et al. (2017) competitively trains DNNs for difficult RL tasks. Their genetic algorithm exhibits remarkable competitiveness with widely used algorithms in the domain of deep RL challenges. Our proposed transfer learning method utilizes MCTS, a distinct search approach not previously applied to these cases.

There has been prior work on applying search algorithms to transfer learning tasks (Singamsetti et al., 2021; Mahajan & Guzdial, 2022; Doosti & Guzdial, 2023). These approaches have shown success at few-shot transfer learning and have generally found that tree-based optimization approaches like MCTS outperform other search algorithms, such as hill-climbing and beam search (Mahajan & Guzdial, 2022; Doosti & Guzdial, 2023). However, these approaches have not previously been applied in an RL setting.

### 2.3 AUTOMATED CHIP PLACEMENT

The chip placement process involves determining the optimal positioning of various macrocomponents, a task that requires sophisticated multi-objective optimization to balance factors such as wirelength, congestion, and density. Various approaches have been developed to address this problem, including conventional divide-and-conquer (Fiduccia & Mattheyses, 1988) and hierarchical placement strategies (Tsay et al., 1988), as well as advanced deep RL techniques (Mirhoseini et al., 2020). Myung et al. (2023) propose that pre-training networks to extract features from different netlists and utilize them as encoders in the policy network can greatly decrease the duration of training. Moreover, Lai et al. (2022) present an approach for placement using RL. This method utilizes convolutional NNs to learn position, wirelength, and view information for circuit modules, resulting in a rich visual representation. Lee et al. (2024) discuss using diffusion models for chip placement. They developed a NN with interleaved graph convolutions and multi-headed attention layers and also a synthetic data generation algorithm. However, there are certain limitations associated with this approach. In particular, synthetic datasets with shorter wirelengths may result in models exhibiting poorer performance to out-of-distribution circuits. We note that we specifically focus on transfer learning as a way to speed up chip design tasks and, as such, do not compare with these approaches that train from scratch.

Historically, simulated annealing (SA) has been a long-standing method with proven success in chip placement (Sarrafzadeh et al., 2003). SA is a search-based approach but optimizes the chip design directly rather than optimizing a chip design model. But SA shows a significant lack of speed, presents challenges in parallelization, and encounters difficulties adapting to current circuits' growing complexity and size.

The RL-based methodology of Mirhoseini et al. (2020) explores an RL agent trained to optimize the arrangement of chips. In our paper, we build upon the policy and value networks from their study.

To the best of our knowledge, our approach is the first to demonstrate that transferring knowledge from a complex chip design task to a simpler chip design can quickly create models that can generate efficient chip placements with reduced cost values compared to training from scratch and fine-tuning approaches.

## 3 SYSTEM OVERVIEW

This section outlines our method, which is divided into two main components. First, we train a model on a source chip design task. After that, our proposed transfer learning method based on MCTS directly modifies the parameters of the trained source model through interactions with the target chip design environment. This method creates a tree of child nodes, each of which represents a distinct model. Lastly, we return the best model, which has the lowest cost value, from the tree.

### 3.0.1 STEP 1: SOURCE TRAINING

The first step involves training a model on the source chip design task $Task_S$ as described in step 1 of Algorithm 1. We apply a specific training configuration for each chip design task due to differences in the numbers of horizontal and vertical routes per micron, as well as the allocation values for macro horizontal and macro vertical routing. We employ an existing architecture of policy and value networks and all other domain-specific hyperparameters from Mirhoseini et al. (2020). The parameters of the policy model are updated using PPO (Schulman et al., 2017) based on the source chip design task $Task_S$. We use an existing proxy cost function from Cheng et al. (2023) as shown in Equation 3.

### 3.0.2 STEP 2: PARAMETER SEARCH

During the second step, we utilize parameter search as a form of transfer learning to adapt the source model to the target chip design task. Initially, the root node $N_R$ corresponds to the source model $S_m$. From this root node $N_R$, we generate a child node by applying a randomly selected search operator to modify its parameters. Child nodes are created until the tree $T$ has reached its maximum number of nodes, $N_M$ which can be specified. To balance exploration with exploitation, we employ an $\epsilon$ value of 0.1, so the best-performing child nodes are selected ninety percent of the time based on the $Cost$ value calculated on the target chip design task $Task_T$ by using the proxy cost function as shown in Equation 3, but less-explored nodes are chosen periodically from all the nodes $N_i$. The selected child node $N_c$ will function as the new root node $N_R$. The node $N_R$ will later go through further parameter modification using a randomly selected search operator to generate a child node $N_j$, calculate the $Cost$ value for the target chip design $Task_T$, and backpropagate this information up the tree. These operations will terminate when the node reaches its maximum length of child nodes $L$. The search operator directly modifies the parameters of the model, allowing a potentially more efficient optimization process, as these can be understood as sparse updates. Lastly, the algorithm outputs the most efficient node $N_{LC}$ based on the $Cost$ value from the search tree $T$ for the target chip design task $Task_T$.

### 3.1 SEARCH OPERATORS

The formulation of our search operators is an essential component of our methodology since these functions directly determine potential sparse parameter modifications during the optimization process. The operators contain two distinct functions that modify the parameter values of the models, with only one operation being randomly selected at a time. These are similar to the mutation functions from prior search-based transfer learning work Singamsetti et al. (2021). We selected these operations because they are simple and systematic, providing a comprehensive range of options to explore the search space. We note that the operators can be applied sequentially to the same layer, allowing for arbitrary value changes in theory. The search operators are as follows:

- The first operation randomly selects a specific layer, $L_r \in \{L_1, L_2, \ldots, L_n\}$, of the policy network $\pi_\theta(a|s)$, where $r \sim \mathcal{U}\{1, 2, \ldots, n\}$ and modifies the weights by adding a scalar value, $a$, uniformly distributed within the range (-1, 1) as shown in Equation (1).

$$W'_{i,j} = W_{i,j} + a, \quad a \sim \mathcal{U}(-1, 1) \tag{1}$$

---

**Algorithm 1** PMCTS Algorithm

---

**Input**: Source Chip Design Task $Task_S$
**Output**: Least cost node $N_{LC}$ for Target Chip Task $Task_T$

1: **Step 1: Source Training**
2: $S_m$=PPO($Task_S$)
3: Save $S_m$
4: **Step 2: Parameter Search**
5: Initialize $S_m$ as root node $N_R$
6: **while** $N_M$ not met **do**
7:     $N_c$ = Selection($N_i$)
8:     $N_R \leftarrow N_c$
9:     **for** step $l = 1$ to $L$ **do**
10:       $N_j$ = Search Operator($N_R$)
11:       Cost($N_j, Task_T$)
12:       Backprop($N_j$)
13:     **end for**
14: **end while**
15: Select $N_{LC}$ from tree $T$
16: **return** $N_{LC}$

---

- The second operation randomly chooses an individual layer of the policy network and involves multiplying the weights of the chosen layer by a scalar value uniformly distributed across the interval [0, 1], as shown in Equation (2).

$$W'_{i,j} = W_{i,j} \cdot a, \quad a \sim \mathcal{U}[0,1] \tag{2}$$

## 3.2 Cost Function

The cost function is identical to the proxy cost calculation function developed by Cheng et al. (2023). It is represented by the *Proxy Cost* estimation, which is the weighted sum of wirelength, congestion, and density costs of the chip design task, as depicted in Equation 3. Domain experts identify these three variables as the most important variables in determining the effectiveness of a specific chip design. The parameters $\gamma$ and $\lambda$ are set to 0.5 in our experiments, as specified by Cheng et al. (2023).

$$Proxy\ Cost = Wirelength\ Cost + \gamma \cdot Density\ Cost + \lambda \cdot Congestion\ Cost \tag{3}$$

The *Proxy Cost* is very essential as it directly influences the learning of the RL agent. The *Wirelength Cost* is the first element of the *Proxy Cost* equation, and it is the normalized half perimeter wirelength (HPWL) shown in Equation 4 where *width* and *height* are the width and height of the chip canvas, respectively, and the *weight*, set to 1 by default, specified by the source pin.

$$Wirelength\ Cost = \frac{1}{|nets|} \sum_{net} \frac{net.weight \times HPWL(net)}{width + height} \tag{4}$$

Secondly, according to the Equation 5, the *Density Cost* is the mean density of the top 10 % most dense grid cells, where *n* is the number of grid cells in the top 10%.

$$Density\ Cost = \frac{1}{n} \sum_{i=1}^{n} Density\ of\ the\ top\ 10\%\ densest\ grid\ cells \tag{5}$$

Lastly, the *Congestion Cost* is divided into two components: congestion resulting from macros and congestion led by net routing. Macro congestion results from the extra routing layer resources used by macros, while routing congestion is caused by the use of routing resources by each routed network. In the process of calculating macro congestion, the horizontal and vertical values of a grid

cell are obtained by summing the routing resources utilized by macros that cross the right and top boundaries of the cell. The total congestion for each grid cell is calculated by summing the macro congestion and routing congestion for each direction individually, as shown in Equation 6. Next, the *Congestion Cost* is calculated as the mean of the top 10% of all $H\_cong$ and $V\_cong$ values of grid cells in the chip canvas defined by the Equation 7. A detailed overview can be found in the prior work Cheng et al. (2023) and the *Circuit Training* repository Guadarrama et al. (2021).

$$H\_cong = H\_macro\_cong + H\_net\_cong$$
$$V\_cong = V\_macro\_cong + V\_net\_cong \tag{6}$$

$$Congestion\ Cost = \frac{1}{n} \sum_{i \in \text{Top 10\%}}^{n} (H_{\text{cong},i} + V_{\text{cong},i}) \tag{7}$$

## 4 EXPERIMENTAL SETUP

In this paper, we focus on the application of our proposed methodology across different chip design tasks. We evaluate our approach on well-understood tasks that have been studied in prior transfer learning work of Mirhoseini et al. (2020). This setting allows us to demonstrate that our method can produce optimized models in less time than corresponding baselines, which typically rely on traditional backpropagation.

### 4.1 ENVIRONMENTS

We examine our approach on four distinct chip designs. The first design refers to the Ariane RISC-V CPU found in the Google Circuit Training repository (Mirhoseini et al., 2020). It consists of 133 hard macro units and serves as a standard reference for the chip placement problem. Hard macros refer to pre-built building blocks (Lavin et al., 2013). The second design, Ariane133, is a slight variant of the first design while maintaining the 133-macro unit configuration (Cheng et al., 2023). In addition, we assess our approach using two designs from the ICCAD04 benchmark, namely IBM01 and IBM02, which consist of 246 and 271 hard macro units, respectively (Cheng et al., 2023; Adya et al., 2004; Adya & Markov, 2002). The complexity of the IBM variations exceeds that of the Ariane versions.

### 4.2 BASELINES

In this research, we utilize two primary baselines: the scratch model and the fine-tuning approach from the prior work of Mirhoseini et al. (2020). Both baselines utilize conventional gradient descent techniques for optimization, providing a comparison to evaluate the efficacy of our proposed method. The training from scratch model baseline is used as a reference for how well an RL agent can perform when trained on the target domain from random initialization. The fine-tuning baseline instead takes a model trained on a source domain and fine-tunes it on a target domain. Through the comparison with these baselines, we show the effectiveness of our approach in creating optimized models for chip placement without primarily depending on gradient descent methods.

### 4.3 COMPUTE RESOURCES

All experiments are conducted using the cloud computing resource provided by < Redacted for Anonymity >, which consists of 18 CPU cores and 2xNVIDIA v100l GPUs with 32GB of memory each. For both training from scratch and fine-tuning approaches, we ran experiments for 12 hours to allow for convergence. In contrast, our PMCTS approach typically runs for shorter durations, ranging from 2 to 8 hours depending on the complexity of the source and target chip designs. For example, using Ariane as the source and Ariane133 as the target chip design, PMCTS completed in 50 minutes with a *max nodes* value of 100, meaning the tree generated 100 nodes before stopping. However, when using Ariane as the source and IBM02 as the target, PMCTS required 2.5 hours to complete due to the complexity of IBM02, where generating new models and calculating the cost for such complex chip designs takes longer. We employ three arbitrary seed values for all experiments, which we give in full in the Appendix A.

| Method | Source | | |
|---|---|---|---|
| | IBM02 | IBM01 | Ariane133 |
| Scratch (Ariane) | 1.29±0.02 | | |
| Fine Tuning | 1.27±0.03 | 1.30±0.05 | 1.28±0.05 |
| Zero Shot | 1.36±0.03 | 1.35±0.00 | 1.32±0.01 |
| Our Approach | **1.19±0.01** | **1.17±0.04** | **1.24±0.07** |

Table 1: Estimated average cost value and standard deviations for the baselines and our approach: when transferring from complex to simple chip (Ariane) designs.

## 4.4 EXPERIMENTS

We evaluate the performance of our approach utilizing four separate chip design tasks. We use all permutations of source and target domains. For example, when assessing our approach's transfer learning performance, we use the Araine chip design task as the source domain to train a model. This trained model is then used as the source input model for our approach, and then we evaluate the performance of the output models from our approach across the three target chip design tasks, namely Ariane133, IBM01, and IBM02. We then repeat this process for each of the other chip design tasks. In addition, for the fine-tuning approach, we take a model trained on a source chip design task and fine-tune it on the remaining target chip designs. We take all our architectures for both scratch and fine-tuning using similar model settings from the same prior study Mirhoseini et al. (2020). For training from scratch and fine-tuning, we execute the models for 12 hours. For our suggested approach, we only consider models from the first hour of running PMCTS. We note that it is very likely, based on prior work, that both from scratch and fine-tuning approaches would eventually outperform the PMCTS models based on the results of prior work of Mirhoseini et al. (2020), but we are focused on speeding up the optimization process, thereby emphasizing this lower timeframe setting.

## 5 RESULTS

In this section, we present our results. The tables show the results obtained by comparing the baseline methodologies with our PMCTS approach. The figures illustrate the average and combined normalized cost values for all relevant approaches, where normalization was performed by dividing each cost value by the maximum cost value. Regarding the figures, our method avoids retraining the same model repeatedly. Instead, it iteratively generates new models by modifying the layers' weights through mutation functions, evaluating the cost value for each newly created model. This iterative process introduces variations in the cost values over time, which explains the observed trends in the figures. The shaded regions of the figures represent the 95% confidence interval range. We separate the results into two categories—one for transferring knowledge from complex to simple chip designs and the other for transferring from simple to complex designs—to clarify the analysis. This splitting allows focused comparisons by emphasizing the differences in performance between the two directions of knowledge transfer. The approach is considered to be better when the cost value is lower. The detailed cost values for each seed corresponding to each method are provided in Appendix A.

Table 1 shows a comparison of the average cost values and standard deviations for the three seed values across four different methods: scratch training, fine-tuning, zero shot and our proposed approach for transferring knowledge from complex source chip designs to the simpler target Ariane chip design task. For the scratch training approach, training the model from scratch on the Ariane chip design results in comparatively higher costs, indicating that the method is unable to identify efficient models within a certain amount of time. For fine-tuning, zero shot and our proposed approach, we use separate source models trained on IBM02, IBM01, and Ariane133 chip designs and then transfer the knowledge to the target Ariane chip design. Fine-tuning offers some improvements over training from scratch. In contrast, our proposed approach consistently outperforms scratch training, fine-tuning and zero shot, resulting in models with lower cost values in all scenarios in shorter times. The results show the usefulness of our method, establishing it as the most cost-effective method for quickly creating efficient models for chip placement tasks.

| Method | Source | |
|---|---|---|
| | IBM02 | IBM01 |
| Scratch (Ariane133) | 1.32±0.04 | |
| Fine Tuning | 1.27±0.02 | 1.34±0.06 |
| Zero Shot | 1.33±0.06 | 1.41±0.06 |
| Our Approach | **1.24±0.03** | **1.27±0.02** |

Table 2: Estimated average cost value and standard deviations for the baselines and our approach: when transferring from complex to simple chip (Ariane133) designs.

| Method | Source | |
|---|---|---|
| | Ariane133 | Ariane |
| Scratch (IBM01) | 2.18±0.08 | |
| Fine Tuning | 2.23±0.15 | 2.25±0.11 |
| Zero Shot | 2.35±0.07 | 2.34±0.06 |
| Our Approach | **2.1±0.05** | 2.23±0.11 |

Table 3: Estimated average cost value and standard deviations for the baselines and our approach: when transferring from simple to complex chip (IBM01) designs.

| Method | Source | | |
|---|---|---|---|
| | IBM01 | Ariane133 | Ariane |
| Scratch (IBM02) | 2.72±0.21 | | |
| Fine Tuning | 2.92±0.26 | **2.27±0.25** | 2.47±0.12 |
| Zero Shot | 2.59±0.15 | 2.37±0.10 | 2.61±0.17 |
| Our Approach | **2.32±0.26** | 2.44±0.08 | **2.43±0.09** |

Table 4: Estimated average cost value and standard deviations for the baselines and our approach: when transferring from simple to complex chip (IBM02) designs.

Table 2 demonstrates the efficiency of our approach for transferring knowledge while using Ariane133 as the target chip design task. As table 2 shows the values for transferring knowledge from the complex chip design tasks, namely IBM02 and IBM01, to the simple chip design, Ariane133. We do not include Ariane as the source model because it is not more complex than the Arian133. When using IBM02 and IBM01 as source designs, our method consistently outperforms baseline approaches, achieving the lowest costs. These results further demonstrate the robustness and efficiency of our method compared to the baselines for complex-to-simple transfer learning tasks.

Tables 3 and 4 present the cost comparisons and standard deviations for knowledge transfer from simpler chip designs to complex ones. In Table 3, when transferring knowledge from Ariane133 and Ariane to IBM01, our approach outperforms fine-tuning but is unable to outperform training from scratch. In Table 3, we leave out IBM02 as the source model for knowledge transfer to IBM01, as IBM02 is not simpler than IBM01. Our approach demonstrates superior performance when using IBM02 as the target chip design task and IBM01 as the source chip shown in Table 4, consistently outperforming scratch training and fine-tuning approaches. Despite these exceptions, our method demonstrates significant overall performance. The experiments indicate there may be a certain level of similarity between IBM02 and IBM01, resulting in the performance of our approach exceeding both initial training and fine-tuning.

Figure 1 illustrates the combined and normalized average cost values of scratch training for the four chip design tasks, fine-tuning approaches for the complex to simpler chip designs, and our proposed method for the complex to simpler chip designs (e.g., using IBM02 as the source and Ariane as the target chip design, etc.). The key focus is on the first hour of training, where our method consistently outperforms both scratch training and fine-tuning in terms of speed and efficiency. The graph demonstrates that our approach quickly identifies the best model, outperforming fine-tuning and scratch training in the same time frame. Furthermore, our method maintains a lower range of normalized cost values throughout the entire period, highlighting its rapid convergence. The faster detection of the best models highlights the efficiency of our approach compared to conventional methods.

Figure 2 displays the combined and normalized average cost values for the four chip designs when training from scratch. It also includes the fine-tuning approaches for simple-to-complex chip designs and our proposed method for simple-to-complex chip designs, such as using Ariane as the source chip design and IBM02 as the target chip design. We observe that our approach encounters challenges when transferring knowledge from simple chip designs to more complex chip designs. Although it does not consistently surpass scratch training and fine-tuning in later time frames, it avoids falling considerably behind.

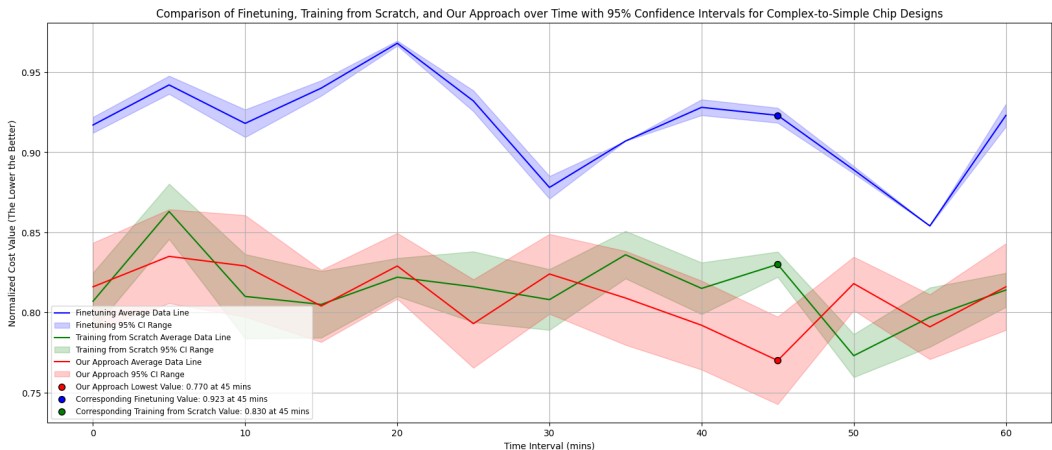

Figure 1: Showing the combined and normalized averaged cost value for all training from scratch models, as well as the complex-to-simple chip design approach of fine-tuning and our proposed method

.

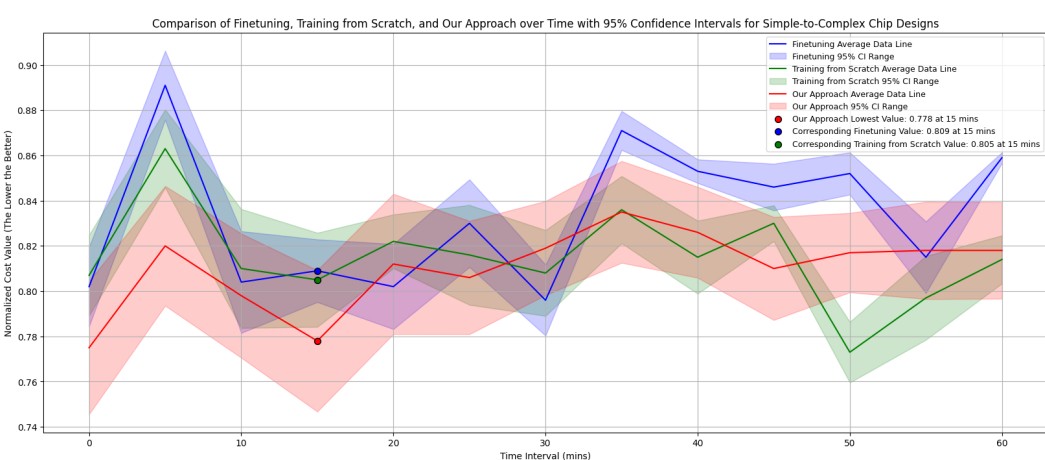

Figure 2: Showing the combined and normalized averaged cost value for all training from scratch models, as well as the simple-to-complex chip design approach of fine-tuning and our proposed method

.

## 6 DISCUSSION

Our proposed transfer learning approach, PMCTS, offers several key advantages that make it highly effective for solving the chip placement problem. One of the primary benefits is its simplicity, as it directly modifies the network's parameters without requiring complex architectural changes or additional training on the target chip domains. Our experiments show that traditional methods such as training from scratch or fine-tuning are time-consuming to reach high-performing models. In contrast, our approach demonstrates greater speed and efficiency, resulting in reduced time and computational costs. This makes it particularly appealing for chip placements where rapid development and cost savings are critical. The shorter period required to generate effective models of our approach, particularly when transferring knowledge from complex to simple chip designs, offers significant advantages, thereby presenting a more economical and scalable solution to the chip industry.

However, it is crucial to acknowledge that this benefit does not apply when the knowledge transfer is reversed, going from simpler to more complex designs. This suggests that the complexity inherent in certain chip designs contains vital information that greatly helps the transfer learning process. However, simpler designs lack sufficient depth to transfer knowledge to complex designs. In the majority of cases, our PMCTS approach shows higher efficiency compared to both training from scratch and fine-tuning approaches. It would be worthwhile to explore the possibility of using either the scratch model or the fine-tuned model as input to PMCTS to further enhance its performance. However, considering that one key advantage of PMCTS is the ability to obtain a more efficient model with significantly less computation time, we believe that this would contradict the purpose of this research. Therefore, we prefer to conduct an additional study of this in the future.

We believe that part of our results, which demonstrate superior performance in transferring knowledge from complex to simple chip designs compared to the reverse, may be due to our selection of search operators. Particularly, our second search operator essentially has the effect of reducing parameter magnitudes in the network. This may align with pruning or other sparse optimization approaches in terms of simplifying a given model, making it more general, and encompassing a simpler task. In Table 9 in the Appendix A, we analyzed the sensitivity of PMCTS to our mutation functions, demonstrating the role of the multiplication operation. When transferring knowledge from complex to simple chip designs, removing the multiplication operator led to quicker convergence to better nodes. For instance, in the IBM01 to Ariane transfer, the minimum cost value of 1.1930 was achieved after creating 30 nodes with all functions, whereas removing the multiplication operator reduced this to just 17 nodes. In contrast, removing the multiplication operator significantly reduced performance when transferring knowledge from simple to complex chip designs. These insights will inform future refinements for our search operators to optimize performance in diverse transfer learning settings. In future work, we hope to explore alternative search operators, which may improve the robustness of our approach.

We plan to extend our approach to more complex chip design tasks, like BlackParrot (Quad-Core) and MemPool Group, to validate the effective trend of complex-to-simple knowledge transfer (Petrisko et al., 2020; Riedel et al., 2023). In addition, alternative search algorithms like Quality-Diversity can be explored given how they offer a wide range of optimized solutions that vary based on a number of user-specified parameters of interest (Chatzilygeroudis et al., 2021).

# 7 CONCLUSIONS

This paper presents a novel transfer learning approach named Parameter Monte Carlo Tree Search (PMCTS) that utilizes MCTS to optimize model parameters for chip placement to generate efficient models in a short timeframe while transferring knowledge from complex chip design tasks to less complex ones and without requiring additional authored transfer features. The speed, along with its effectiveness, makes our approach potentially useful for chip placement scenarios where rapid optimization and implementation are crucial. By conducting experiments on four different chip design tasks—Ariane, Ariane133, IBM01, and IBM02—we demonstrate that knowledge transfer from complex chip designs to less complex ones results in improved performance, surpassing the training from scratch and fine-tuning in terms of both time and performance. In contrast, the transfer from simpler to more complex designs does not produce the same advantage in all scenarios.

# 8 ETHICS STATEMENT

We identify that our work was done in partnership with an international technology company involved in chip manufacturing. While our partner hopes to benefit from this research, we will make all of our code accessible to the public, and we utilize publicly available chip design environments for evaluation. We also note that while job loss is possible with any AI application, presently this type of automated chip design task is handled via search-based approaches in industry. As such, we do not identify any negative potential repercussions in terms of job loss from our approach.

## 9 REPRODUCIBILITY STATEMENT

For reproducibility, we make use of only publicly available chip design task environments as covered in Subsection 4.1. We also make use of previously published network architectures and optimization approaches as covered in Subsection 4.4. Finally, we include all code in the supplementary materials and identify the seeds used for all experiments in our Appendix A.

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

## A  APPENDIX

Here we give the results of the experiments in detail, with one table for each of the source target tasks.

| Method | Source | Cost (Seed 10) | Cost (Seed 55) | Cost (Seed 111) |
|---|---|---|---|---|
| Scratch (Ariane) | | 1.312 | 1.303 | 1.268 |
| Fine Tuning | IBM02 | 1.273 | 1.296 | 1.232 |
| Our Approach | | **1.202** | **1.176** | **1.193** |
| Fine Tuning | IBM01 | 1.267 | 1.267 | 1.360 |
| Our Approach | | **1.193** | **1.132** | **1.193** |
| Fine Tuning | Ariane133 | 1.245 | 1.333 | 1.268 |
| Our Approach | | **1.193** | **1.312** | **1.202** |

Table 5: Estimated cost for baselines and our approach: transfer learning from complex to simple chip (Ariane) designs.

| Method | Source | Cost (Seed 10) | Cost (Seed 55) | Cost (Seed 111) |
|---|---|---|---|---|
| Scratch (Ariane133) | | 1.347 | 1.352 | 1.272 |
| Fine Tuning | IBM02 | 1.27 | 1.255 | 1.289 |
| Our Approach | | **1.245** | **1.206** | **1.264** |
| Fine Tuning | IBM01 | 1.272 | 1.352 | 1.389 |
| Our Approach | | **1.265** | **1.266** | **1.292** |

Table 6: Estimated cost for baselines and our approach: transfer learning from complex to simple chip (Ariane133) designs.

| Method | Source | Cost (Seed 10) | Cost (Seed 55) | Cost (Seed 111) |
|--------|--------|----------------|----------------|-----------------|
| Scratch (IBM01) | | 2.222 | 2.233 | 2.096 |
| Fine Tuning | Ariane | 2.343 | **2.133** | **2.264** |
| Our Approach | | **2.194** | 2.145 | 2.359 |
| Fine Tuning | Ariane133 | 2.085 | 2.381 | 2.216 |
| Our Approach | | **2.039** | **2.145** | **2.101** |

Table 7: Estimated cost for baselines and our approach: transfer learning from simple to complex chip (IBM01) designs.

| Method | Source | Cost (Seed 10) | Cost (Seed 55) | Cost (Seed 111) |
|--------|--------|----------------|----------------|-----------------|
| Scratch (IBM02) | | 2.932 | 2.504 | 2.714 |
| Fine Tuning | Ariane | **2.324** | 2.551 | 2.526 |
| Our Approach | | 2.397 | **2.526** | **2.364** |
| Fine Tuning | Ariane133 | **2.266** | 2.532 | **2.025** |
| Our Approach | | 2.360 | **2.526** | 2.444 |
| Fine Tuning | IBM01 | 2.704 | 2.84 | 3.211 |
| Our Approach | | **2.537** | **2.028** | **2.390** |

Table 8: Estimated cost for baselines and our approach: transfer learning from simple to complex chip (IBM02) designs.

| Source | Target | Operation Type | Min Cost Value | Min Nodes |
|--------|--------|----------------|----------------|-----------|
| IBM01 | Ariane | All Operations | 1.1930 | 30 |
| | | Without Multiply | 1.1930 | 17 |
| IBM02 | Ariane | All Operations | 1.1930 | 164 |
| | | Without Multiply | 1.1930 | 111 |
| Ariane133 | IBM02 | All Operations | 2.4436 | 88 |
| | | Without Multiply | 3.270 | 37 |
| Ariane133 | IBM01 | All Operations | 2.101 | 109 |
| | | Without Multiply | 2.347 | 79 |

Table 9: Sensitivity Analysis of the Search Operators of the PMCTS Approach.

