# OpenReview forum: "Parameter Monte Carlo Tree Search: Efficient Chip Placement via Transfer Learning"
_ICLR.cc/2025/Conference — Submitted to ICLR 2025_

### Official Review · Reviewer_4sxi · 2024-11-02

**Soundness:** 3
**Presentation:** 2
**Contribution:** 2
**Rating:** 3
**Confidence:** 2

**Summary:**

This paper presents a new transfer learning method in chip design called Parameter Monte Carlo Tree Search (PMCTS). It leverages Monte Carlo Tree Search (MCTS) to transfer knowledge from deep reinforcement learning (RL) models trained on one chip design task to another. By searching directly over the model parameters, it generates models for efficient chip placement.

**Strengths:**

1. Efficient chip placement can be achieved through transfer learning algorithms, and complex chip designs can be transferred to simpler chips.

2. The proposed PMCTS method has successful applications and outperforms other transfer learning methods across multiple chip designs tasks, but the paper does not seem ready for publication.

**Weaknesses:**

1. The paper lacks images to clearly illustrate the proposed method, making it difficult for readers to understand.

2. The paper lacks necessary ablation studies on the cost function, and it is important to discuss the impact of different weights $\gamma$ and $\beta$ on the results.

3. The paper does not provide a thorough discussion of the limitations of the proposed method.

4. The PPO used in the method lacks the introduction of necessary preliminaries for reinforcement learning, such as the definitions of MDP, state, and action.

5. Monte Carlo Tree Search requires significant computational resources. The authors mention the performance advantages compared to the baseline but do not address the issue of computational complexity. It is necessary to quantitatively compare the computational cost and GPU usage of the proposed method with the baseline.

6. The paper lacks necessary design illustrations, and they are not provided even in the appendix.

7. The authors need to provide more detailed information about the training process, such as the number and distribution characteristics of the training and test samples.

8. The authors did not provide the network architecture and training parameters. Presenting these in tables within the paper would allow readers to better understand the network's composition.

**Questions:**

Please check the weaknesses.

---

> ### Author Response · Authors · 2024-11-27
> **Response to Reviewer 4sxi**
>
> We appreciate the feedback and acknowledge the points raised. We are actively working to address several of these issues and will incorporate the necessary updates in the revised manuscript. We would like to clarify that our MCTS approach has access to the same amount of computation compared to our fine-tuning baseline.

---

### Official Review · Reviewer_HTY2 · 2024-11-04

**Soundness:** 2
**Presentation:** 3
**Contribution:** 1
**Rating:** 3
**Confidence:** 3

**Summary:**

This paper proposes a method to finetune a pretrained policy on chip designs to new tasks by employing MCTS on policy parameters. By perturbing certain layers with operators as addition and multiplication with random noise and applying MCTS, the resulting policy can be adapted faster when few time is allowed compared to from-scratch pretraining on finetuning. The proposed method shows promising results when adapting from complex to simpler problems while it underperforms classical finetuning when trying to generalize from simple to larger tasks.

**Strengths:**

The writing is clear, and several relevant related works are mentioned. The proposed method is simple enough to be adaptable to a wider range of tasks than just fine-tuning placement policies for chip design. Source code is provided, which is appreciated.

**Weaknesses:**

This paper raises several concerns in the current state.

1. I believe the novelty is pretty limited. In my understanding, in terms of modeling, problem formulation, etc, the method is no different from Mirhoseini et al. (2020), for instance, Section 3.2 and 3.0.1.  The main contribution is the application of MCTS to the parameter search operators proposed in Singamsetti et al. (2021) and to chip design.I believe the proposed method could benefit from the further evaluation on different domains or other problems that involve finetuning in the same domain, as I do not see a lack of generality per se -- if the method is proven to work in other (toy) example problems, it would greatly strengthen the paper.

2. This is recognized by the authors -- the paper does not improve the performance against finetuning when generalizing from simple to complex designs. Although this is a weakness, I appreciate the authors in being straightforward.

3. The authors propose two search operators to randomly perturb model layers, i.e. adding and multiplying uniformly distributed noise. However, there is no ablation study about this or sufficient analysis on sensitivity to the operator choice.

4. I am not convinced by the performance shown in Figures 1 and 2. Namely, one would expect the performance to improve over time. However, it appears to me as if the performance is almost random (no clear trend over time). Moreover, performance actually gets much worse over time in terms of cost, as shown in Figure 2. How is this possible? Also - the authors should include zero-shot performance to Tables 1 to 4. Otherwise, it is hard to assess the improvement.

**Questions:**

1. When does finetuning become better than the proposed method? You mentioned ““We note that it is very likely, based on prior work, that both from scratch and fine-tuning approaches would eventually outperform the PMCTS models based on the results of prior work of Mirhoseini et al. (2020), but we are focused on speeding up the optimization process, thereby emphasizing this lower timeframe setting.”. I wonder what would be the convergence point in which finetuning is better.

2. Would it be possible to use PMCTS in combination with finetuning?

3. Which other tasks in the broader ML for optimization community could your method expand to?

4. What is the effect of search operators? I.e., if you just used the second one, would it perform as well?

---

> ### Author Response · Authors · 2024-11-27
> **Response to Reviewer HTY2**
>
> We appreciate the insightful feedback you provided. We would like to respond to a number of things that you mentioned.
>
> 1. Thank you for the feedback. We have conducted additional experiments on well-established transfer learning benchmarks: CIFAR-10 and CIFAR-100. Kindly refer to our response to Reviewer VXCa, Point 1, for results and analysis.
> 2. Thank you for pointing this out the need for ablation studies to analyze the sensitivity of our approach to the choice of search operators. This data will be included in the next draft of our manuscript.
> 3. Regarding the figures, our method takes a distinct approach by avoiding retraining the same model repeatedly. Instead, it iteratively generates new models by modifying the weights of layers through mutation functions, evaluating the cost value for each newly created model. This iterative process introduces variations in the cost values over time, which explains the observed trends in the figures. While this may appear as a lack of consistent improvement, it reflects the exploratory nature of PMCTS in navigating the solution space. Additionally, we will include zero-shot performance data in Tables 1 to 4 in the updated manuscript to provide a clearer assessment of improvements. Thank you for your valuable feedback.
> 4. Thank you for your question regarding when finetuning becomes better than the proposed method. Please refer to our response to Reviewer VXCa, Point 3, where we address this concern.
> 5. Yes, it is possible to use PMCTS in combination with fine-tuning. This hybrid approach could leverage PMCTS to explore diverse model configurations through its mutation-based optimization while subsequently applying fine-tuning to refine the best-performing model. We consider this an interesting direction for future work, but we lack sufficient time to complete the experiments before the rebuttal deadline.

---

### Official Review · Reviewer_V9QD · 2024-11-04

**Soundness:** 2
**Presentation:** 1
**Contribution:** 1
**Rating:** 3
**Confidence:** 4

**Summary:**

This paper presents a transfer learning approach known as Parameter Monte Carlo Tree Search (PMCTS), aimed at optimizing model parameters for chip placement, thereby enhancing efficiency and reducing time. PMCTS exhibits superior performance and faster optimization compared to both training from scratch and fine-tuning methods. Experimental results across various chip design tasks indicate that the approach significantly enhances design efficiency, especially in situations involving knowledge transfer from complex to simpler designs.

**Strengths:**

The idea of transferring knowledge through MCTS in chip design is good.

The proposed approach demonstrates better performance than fine-tuning and training from scratch, which makes sense to me.

**Weaknesses:**

The quality of the figures is very poor. Please increase the font size and reduce the size of the figures.

The types of chips tested in the experiments are very limited. It is recommended to evaluate the results on popular benchmarks such as ISPD2005 and ICCAD2015.

The number of baseline methods for comparison is too few to assess the performance of this approach in the current chip placement domain.

Please check and revise the formatting of the references.

**Questions:**

None.

---

> ### Author Response · Authors · 2024-11-27
> **Response to Reviewer V9QD**
>
> We appreciate the insightful feedback you provided.  We would like to respond to a number of things that you mentioned.
>
> 1. We appreciate the feedback regarding the quality of the figures and their readability. In the revised version of the manuscript, we will increase the font size of all labels to ensure they are easily readable.
> 2. We appreciate the feedback regarding the limited range of chips tested in our experiments. Due to time constraints, we could not complete these experiments prior to the rebuttal deadline. We acknowledge the importance of these benchmarks for a comprehensive evaluation and are actively working on conducting additional experiments by converting the benchmark designs used in the ChipFormer into proto-buff formats because the IBM and Ariane designs that we used from the Google circuit training were in proto-buff formats.
> 3. We appreciate the concern regarding the limited number of baseline methods used for comparison in the chip placement domain. As mentioned in our response to Reviewer VXCa, Point 2, we have conducted experiments with zero-shot learning and demonstrated that our approach outperforms it across multiple transfer tasks. However, we acknowledge the need to evaluate our method against additional baseline approaches for a more comprehensive assessment.

---

### Official Review · Reviewer_dcPZ · 2024-11-04

**Soundness:** 2
**Presentation:** 1
**Contribution:** 2
**Rating:** 3
**Confidence:** 3

**Summary:**

This paper proposes a prior search-based transfer learning method for chip placement, in which pre-trained network parameters are updated via Monte Carlo Tree Search (MCTS). The proposed method enables cost-effective adaptation for chip placement. The authors validated the method on four different problems, comparing it to the full-training and fine-tuning approaches.

**Strengths:**

1. The proposed method directly updates network parameters through search, achieving improved time-performance.

2. The paper clearly showed the related works and explained their differences and limitations.

**Weaknesses:**

1. Lack of Background Information: There is insufficient background information regarding the objective of the target task. While the method aims to transfer knowledge from complex chip design tasks to simpler ones, it is unclear how practical this approach is given the increasing complexity of chip design, with more design components and constraints.

2. Limited Performance in Simple-to-Complex Transfer Learning: The performance of the method in transferring from simple to complex tasks is not promising. According to Line 341, "both from-scratch and fine-tuning approaches would eventually outperform the PMCTS models," indicating only a fast initial improvement. Additionally, Figures 1 and 2 suggest that the model may not be learning effectively, as costs do not consistently decrease. In Figure 2, the proposed method even concludes with a higher cost than its starting point.

3. Lack of Ablation Studies: The paper lacks ablation studies, particularly regarding the search method for parameter updates and the selection of hyperparameters.

4. Unclear Tables and Figures: The tables lack informativeness, and the figures are difficult to interpret. Adjusting the label sizes on figures would improve readability. Additionally, adding a column to the tables showing the time cost for each method would allow a clearer evaluation of time-performance across methods.

**Questions:**

1. Could you conduct an ablation study on the percentage of selecting the best-performing child node versus selecting a less-explored node, and report its effect on performance?

2. Is this method applicable to other domains?

---

> ### Author Response · Authors · 2024-11-27
> **Response to Reviewer dcPZ**
>
> We appreciate the insightful feedback you have provided.  We would like to respond to a number of things that you mentioned.
>
>
> 1. Thank you for pointing out the limitations in transferring knowledge from simpler to more complex tasks. We agree that the performance of our method in this specific scenario is not as promising, and this observation forms one of the key contributions of our paper. Specifically, we demonstrate that transferring knowledge from a complex chip design task, such as IBM02, to a simpler chip design, like Ariane, outperforms both training from scratch and fine-tuning approaches in terms of creating chip placements with reduced cost values within shorter timeframes. However, transferring knowledge from a simpler chip design to a more complex one does not yield the same benefits, highlighting the inherent challenges in such cases.
> 2. Regarding Figures 1 and 2, our PMCTS method avoids retraining the same model. Instead, it iteratively generates new models by modifying the weights of layers through mutation functions and calculating the cost value for each newly created model. This iterative mutation process leads to variations in the cost values during the process, which explains the observed trends in the figures.
> 3. We acknowledge the significance of including ablation studies.  We are actively working on conducting comprehensive ablation studies and will incorporate the findings into the next draft of the paper.
> 4. We recognize the importance of readability in presenting results. We will improve the label sizes and overall design of the figures in the next draft of the paper.

---

### Official Review · Reviewer_VXCa · 2024-11-04

**Soundness:** 2
**Presentation:** 2
**Contribution:** 1
**Rating:** 3
**Confidence:** 3

**Summary:**

This paper proposes a new MCTS-based algorithm for parameter search in transfer learning. The proposed algorithm is used in the chip placement task, where a RL model is trained on a source chip circuit and then transfered to a new one. The results show that the proposed method can shorten the time needed in the transfering stage, compared with directly fine-tuning.

**Strengths:**

This paper identifies that transfer learning is an important task in chip placement, and proposes a MCTS-based transfer learning approach. The results on several chip circuits show that the proposed transfer learning approach is faster than fine-tuning approach.

**Weaknesses:**

1. The proposed PMCTS method is designed for general transfer learning task, rether than specifically designed for the chip placement task. If the authors claim that the core contribution of the paper is this general algorithm, they may need to evaluate the algorithm on general transfer learning tasks, rather than only considering chip placement.
1. The authors mainly compare the proposed method with directly fine-tuning. However, other transfer learning method should also be compared with.
1. The performance improvement is not significant. Moreover, since fine-tuning method may outperform transfer learning in a longer time, the authors may want to further extend the training time to see whether transfer learning approach still demonstrates a better performance after converggence.
1. Only limited models and datasets are used for evaluation. For example, ChiPFormer, a pretraining approach, is not considered for comparision.

**Questions:**

See weaknesses.

---

> ### Author Response · Authors · 2024-11-27
> **Response to Reviewer VXCa**
>
> We appreciate the insightful feedback you provided.  We would like to explain a number of things that you mentioned in the weakness part.
>
> 1. We acknowledge your point regarding the need to evaluate PMCTS on general transfer learning tasks rather than limiting its application to chip placement. To address this, we conducted additional experiments on widely recognized benchmarks: CIFAR-10 and CIFAR-100. We ensured consistency across all methods by using the same seed value (42), model architecture (ResNet50 pre-trained on ImageNet), optimizer (Adam), learning rate (0.001), and number of epochs (10). The results demonstrate that our PMCTS approach outperforms training from scratch and fine-tuning methods. Specifically, PMCTS achieved a test accuracy of 79.41% on CIFAR-10 when transferring knowledge from CIFAR-100 and 49.06% on CIFAR-100 when transferring knowledge from CIFAR-10, compared to 39.98% and 8.95% when training from scratch. For the fine-tuning approach, the test accuracy was 73.80% on CIFAR-10, and for CIFAR100 it was 36.28%. These findings demonstrate that PMCTS is not only effective for chip placement but can also perform well on general transfer learning tasks.
> 2. Thank you for this valuable feedback around the need for additional experiments. Due to time constraints, we were unable to complete additional experiments before the rebuttal deadline. However, we have conducted initial evaluations against zero-shot transfer learning, and our proposed PMCTS approach demonstrates superior performance. Specifically, for the Ariane133 to Ariane transfer, zero-shot achieved a cost value of 1.334, whereas our approach obtained 1.201. Similarly, for Ibm01 to Ariane, zero-shot scored 1.36 compared to our result of 1.193. Lastly, for Ariane to Ariane133, zero-shot reached 1.455, while our approach achieved a lower cost of 1.400.
> 3. We appreciate the suggestion to evaluate performance under prolonged training durations. We ran an experiment across two seeds comparing the performance of fine-tuning and PMCTS with nearly 20 hours of compute. For the Ariane133 to Ariane transfer, fine-tuning achieved an optimum average cost value of 1.141±0.01, whereas our approach resulted in an average cost of 1.197±0.00. In the IBM01 to Ariane transfer, fine-tuning reached a value of 1.229±0.04, and our approach achieved a value of 1.193±0.02, demonstrating improved performance. For the IBM02 to Ariane transfer, fine-tuning yielded an optimum averaged cost value of 1.198±0.03, whereas our approach achieved a similar cost of 1.1975±0.01. Thus we identify that we achieve similar results with sufficient computation. But we note that our approach outperforms finetuning early on. Thus, we can approximate the final performance more quickly with our approach.
> 4. We thank the reviewer for the recommendation, we have begun to explore additional baselines and hope to include those results prior to the end of this discussion.

---

### Author Response · Authors · 2024-11-28
**Note to the Reviewers**

We appreciate the valuable feedback provided, which has helped us strengthen our manuscript. In response to the comments and suggestions, we have made the following updates to the paper:
1) Added information regarding the time required to run the different approaches in the Compute Resources section.
2) Included zero-shot transfer learning cost values in Tables 1 to 4 for a comparison of our PMCTS method.
3) Provided an explanation for the variance in the cost values of our approach in Figures 1 and 2 within the Results section.
4) Added a sensitivity analysis of the search operators of the PMCTS approach as an ablation study in the Appendix section. This study evaluates the impact of different search operators (e.g., removing the multiplication operation) on the performance of our method across both complex-to-simple and simple-to-complex knowledge transfer scenarios.

We hope these additions improve the clarity and quality of the manuscript.

---

### Meta-Review · Area_Chair_58DV · 2024-12-19

**Metareview:**

This paper proposes PMCTS, a Monte Carlo Tree Search-based approach for transferring knowledge in chip placement tasks by directly searching over model parameters. While reviewers acknowledged the paper's attempt to improve transfer learning efficiency in chip design, significant concerns were raised about its limited technical novelty, insufficient experimental validation, and lack of comprehensive comparisons with existing approaches. The authors' responses demonstrated the method's effectiveness on additional transfer learning benchmarks like CIFAR, but the core concerns about the method's performance instability, limited ablation studies, and underwhelming performance on simple-to-complex transfer scenarios remain unaddressed. Additionally, the paper would benefit from clearer presentation of experimental results and better discussion of computational requirements.

Given these limitations and the preliminary nature of the current results, I recommend rejection with encouragement to address these issues in a future submission.

**Additional Comments On Reviewer Discussion:**

While reviewers acknowledged the paper's attempt to improve transfer learning efficiency in chip design, significant concerns were raised about its limited technical novelty, insufficient experimental validation, and lack of comprehensive comparisons with existing approaches. The authors' responses demonstrated the method's effectiveness on additional transfer learning benchmarks like CIFAR, but the core concerns about the method's performance instability, limited ablation studies, and underwhelming performance on simple-to-complex transfer scenarios remain unaddressed.

---

### Decision · Program_Chairs · 2025-01-22

Reject